# Pantograph Sliding Strips Failure—Reliability Assessment and Damage Reduction Method Based on Decision Tree Model

**DOI:** 10.3390/ma14195743

**Published:** 2021-10-01

**Authors:** Małgorzata Kuźnar, Augustyn Lorenc, Grzegorz Kaczor

**Affiliations:** Department of Rail Vehicles and Transport, Faculty of Mechanical Engineering, Cracow University of Technology, 31-155 Kracow, Poland; alorenc@pk.edu.pl (A.L.); gkaczor@pk.edu.pl (G.K.)

**Keywords:** reliability assessment, failure distribution model, pantograph strip, AI methods, machine learning, artificial neural network, damage prevention

## Abstract

Damage to the pantograph or sliding strip may cause the blocking of the railway line. This is the main reason for which the prediction of pantographs’ failure is important for railway carriers and researchers. This article presents a sliding strips failure prediction method as a main means of preventing disruptions to the transport chain. To develop the best predictive model based on the decision tree, the complex tree, medium tree and simple tree machine learning methods were tested. Using a decision tree, the categorization of the given technical conditions can be properly realized. The obtained results showed that the presented model can reduce sliding strip failure by up to 50%. Special attention was paid to the current collector (AKP-4E, 5ZL type), measured during periodic reviews of locomotives EU07 and EU09. To assess the reliability of the selected pantograph strips, a non-destructive degradation analysis was carried out. On the basis of the wear measurements of the strips and the critical value of wear, a failure distribution model was developed. Operational data, collected during periodic technical reviews, were provided by one of the biggest railway carriers in Poland. The results of the performed analyses may be used to build a preventive maintenance strategy to protect pantographs. The applied reliability models of wear propagation can be extended by the parameters of the cost and repair time becoming the basis for estimating the costs of operation and maintenance.

## 1. Introduction

Reliability and safety during the operation of railway vehicles largely depend on the correct power reception from the catenary system by a traction vehicle. The technical condition of a pantograph is checked during every technical review. According to the preventive maintenance strategy related to the pantograph, we distinguish the following activities: Control reviews (every 2–4 days);Periodic reviews (once a month);Large reviews (every 250,000 km ± 10%);Smaller repair (every 500,000 km);Bigger repair (every 1,000,000 km);Major repair (after the course of 4,000,000 km).

During each of the reviews, among others, the visual inspection of a current collector is made which takes into account checking the current collector components without its disassembly.

A component which is in direct contact with the contact wire of an overhead catenary line is a carbon sliding strip. The failure of the strip may result in dangerous and expensive damages to the catenary line. 

Many scientific papers regarding the pantograph–overhead catenary system have been published to date. One of the first papers, which focused on the problem of contact sliding strips and the catenary, was published in 1970 by Abbot [1]. He presented the mathematical model for predicting the dynamic behavior of a trolley wire overhead contact system disturbed by the pantograph on the roof of a train. This area of research was also investigated by Aboshi [2], who presented methods to estimate the static installation quality of the contact lines by calculating their dynamic characteristics and reporting some examples of the estimation of real line conditions. Wąkatroba et al. [3] used the MATLAB Simulink environment to simulate the pantograph behavior. A more detailed model was presented by Wilk et al. [4,5]. They presented two mathematical models of railway current collectors including two degrees of freedom. The first model, henceforth referred to as the pantograph articulated model (PAM), has the first degree of freedom in a rotational motion and the second degree of freedom in a translational motion. The second model, henceforth referred to as the pantograph reference model (PRM), has both degrees of freedom in a translational motion. The differential equations of the PAM contain very complex coefficients dependent on the rotation angles of individual arms. The single mass, dual mass and multimass models were described by Judek et al. [6]. The authors presented the aspects of the pantograph–catenary interaction based on the mathematical models’ comparison. Rusu Anghel et al. [7] proposed a numerical model for the catenary suspension–pantograph assembly. Through the application of this model, a control system was implemented for the contact force in order to improve the current collector’s quality and reduce its energy loss.

Aboshini and Manabe [8] presented a model for establishing a relation between the wave motion of the contact wire and the contact force fluctuation of the pantograph. One of the most important subjects of the overhead contact line and pantograph system is to reduce the contact loss of the pantograph during high-speed operation. The presented model was focused on this problem. Mokrani and Rachid presented a simplified model with three degrees of freedom (3-DOF) of the pantograph–catenary (PAC) using the fuzzy sliding mode controller. Abdullach et al. [9,10] focused on the integration of the catenary model and the pantograph model during the simulation flow of the contact force variations. They proposed a sinusoidal feed forward force and a simple feedback control force used for controlling the wave-like contact force fluctuations by means of active dampers. Piga-Carboni et al. [11] considered in their research the compensation of the equivalent stiffness of the catenary as an uncertainty. They also proposed a robust, nonlinear output–feedback scheme. Makino et al. [12] proposed a model for reducing the vibration and noise of current collectors. Their model allows reducing push-up variation by approximately 67%. Pappalardo et al. [13] proposed a three-dimensional multibody system (MBS) model of a pantograph mounted on a train developed using a nonlinear augmented MBS formulation to protect the catenary from deformation. Pisano and Usai [14] analyzed one of the main problems in high-speed train transportation systems related to the current collection quality, namely the fact that it may be dramatically decreased because of oscillations in the pantograph–catenary system. This problem was addressed by means of active pantographs. In their research, some results concerning the possible implementation of variable structure control (VSC) techniques on a wire-actuated symmetric pantograph were presented.

Yuan et al. [15] focused on the preparation and tribological behavior of carbon fiber-reinforced pantograph sliding strips. Rojek and Majewski [16] focused their attention on the laboratory tests of the new material of contact sliding strips–carbon composite. Tao et al. [17] and the research team of Ding et al. [18] focused on the tribology behaviors of a carbon strip/copper contact wire of a pantograph/centenary system under electric current. Qian et al. [19] presented the development of carbon contact strips and research progress in China. Kubo and Kato [20] presented the effect of the arc discharge on the wear rate and the wear mode transition of the sliding strips. Ding et al. [21] analyzed the impact of the electric arc and temperature in terms of damage caused to the sliding strips and catenary. They performed a series of tests on the friction and wear performance of pure copper rubbing against the carbon strip under electric current. These tests were carried out on a modified pin-on-disc friction and wear tester. Ding et al. [21] and Manory [22] analyzed the copper–graphite composite materials (CGCMs). The wear mechanism of the CGCMs versus Cu was identified as a combination of mechanisms that change in function of variations in the composition. The wear on the contact wire was below a measurable rate, and the wear rate of the CGCM samples against pure copper was very low. They proved the advantages of using this material for applications such as pantographs.

Some of the studied works were mainly concerned with the numerical methods of the simulation of dynamic phenomena [1,2,3,4,5,6,7], the analysis of contact force [8,9,10,11,12,13,14,23] as well as the wear of sliding strip material [15,16,17,18,19,20,21,22,24]. These papers were particularly focused on the material properties depending on the composition of the strip, and the interaction between the sliding strip and catenary. Despite wide interest in the problems of the pantograph–overhead catenary system resulting from the desire to ensure its best cooperation and reduce operating costs, only a small number of studies have been published concerning the technical condition of sliding strips during technical reviews of a pantograph. The problem associated with the correct determination of the technical condition is therefore very important. Kuźnar [25] proposed a method of the trivalent assessment of the technical condition of a rail current collector’s sliding strips on the basis of which the three-valued technical state method used in this article was used. Kuźnar et al. [26] developed a model for reducing the delays in the supply chain by using maintenance decision support systems for the pantograph, proving that neural networks can be used for this purpose. 

A new area of research is that of predictive maintenance. Predictive maintenance uses historical and real-time data to anticipate problems before they happen. In their research, Ren-Hong X. et al. [27] focused on the predictive maintenance of the catenary system. They proposed a decision support model using both mixed integer programming and heuristic methods to identify and assign catenary maintenance tasks to minimize maintenance costs and labor costs. The numerical results of their research show that the cost can be improved by 25% compared to the current PM-only practice. In 2020, Hongwei Z. et al. [28] proposed using predictive maintenance for electric multiple-unit (EMU) trains running at a speed of 250 km/h. They highlighted the technical problems of the current system for high-speed trains in terms of the wheel–rail relationship, the pantograph–catenary relationship, resistance and noise. They claim that predictive maintenance ensures safe and reliable EMU operation and reduces the life-cycle cost.

In this paper, special attention was paid to the tribological wear of the sliding strips of a current collector (AKP-4E and 5ZL type), measured during the periodic reviews of locomotives of types EU07 and EU09. A reliability assessment was also made for a sliding strip and a predictive model based on a decision tree was proposed to prevent the failures of sliding strips.

## 2. Materials and Methods

### 2.1. Damage of Pantograph Sliding Strips

Currently, there are sliding strips made of carbon composite. The detailed technical properties are presented in Table 1.

In Polish companies, the material parameters for this component are defined as:−The weight content of metal in carbon material in the 3 kV DC traction power system is <40%;−The chemical composition (according to the UIC 608: 2003 standard) is C = 75–77%; CU = 12–15%; Pb = 7–10%; Sb = 1–2%

Verification of the technical condition of the sliding is performed during every technical review. At the time of examination, it should be remembered that the sliding strip exchange can be caused by three types of destructive processes: wear and small failures due to tribological wear; failures of the sliding strip; and changes in pantograph mechanical regulations influencing the contact force. The photos presenting damages in Figure 1 and Figure 2 were taken during technical reviews and after the failure of the pantograph during transportation, respectively. In Figure 1a, minor damages are presented. These damages follow from the uneven edge of the sliding strip. In Figure 2b, the uneven edge and material extraction are presented. The uneven edge of the pantograph sliding strips may wear the overhead line more quickly or even damage it.

In the case of wear, a reduction in the thickness of the strip may be noticed as a result of the abrasion processes and electro-eroding phenomena. The wear process induces an approximately monotonous change in the thickness of the sliding strip. The reason for replacing the sliding strip in this case exceeds the recommended strip thickness. If some small defects occur—which do not cause any loss of the strip’s current collection ability, e.g., by wearing the edge of the strip—there is no need to replace the strip (Figure 1a). Such damages are caused by the impact on the hard points of the catenary and it is often assumed that minor surface damages may not exceed 30% of the surface of the carbon strip. However, if there is any major damage (Figure 1b), then the strip should be replaced, because it may damage the overhead line.

In the case of replacement following damage, there are steps to be undertaken following certain evaluation criteria such as:Material melting as a result of arcing and damages caused by arcing (Figure 2a);Detachment of a piece of carbon strip (Figure 2b);Cracks of a sliding strip (Figure 2c);The top layer of a carbon strip is peeling off (Figure 2d).

During the maintenance activities, there are also some changes in pantograph mechanical regulations which can cause the uneven wear of sliding strips. In such a case, if the difference in strip thickness is big, then the strips should be replaced. If the thickness is different in a strip but acceptable, then the slide should be turned 180 degrees.

The cause of damage depends on many correlated factors such as the contact force between the current collector and catenary, air humidity, air temperature, ice on the catenary lines, material defects of the sliding strips and the poor quality of the railway infrastructure. Due to the multi-factor impacts on the failures of the current collector, it is impossible to use typical mathematical modeling and linear programming. To predict the damages, the heuristics method and artificial intelligence method may be applied. In this paper, decision tree machine learning was used. An exact description of the proposed model is presented in the next subsection.

### 2.2. Machine Learning Method for Failure Prediction of the Pantograph Sliding Strips

In order to reduce the number of strip failures, a model based on the machine learning method was applied. In order to develop such a model, it was necessary to process the archival data by the supervised learning method and then select as well as perform an implementation of the best predictive model for simulation. The term “archival data” refers to data that are not actual—these come from past technical reviews. The presented method can also be used in real time. Thus, if the company uses this method, then the results are based on real-time data.

This method is presented in detail in Figure 3 and Figure 4. Figure 3 shows the first step of the analysis in which, in addition to training, a failure analysis was performed according to the archival data (reference variant). The second step, presented in Figure 4, contains the prediction of the technical condition of the current collector and the failure analysis for the data modified in accordance with the prediction results (variant I). Step I also includes the reliability assessment for the further processing of output data.

### 2.3. Data Preparation for the Machine Learning

Reliability assessment and expert knowledge allowed the elaboration of identification algorithms for technical condition and replacement causes. The exemplary algorithms are shown below:(1)$$Wop=1\Leftrightarrow Nl_{i+1}=Nl_{i}\;\wedge (Top_{i+1}\neq \;Top_{i}\;\vee \;Nop_{i+1}\neq \;Nop_{i})$$
(2)$$Wn=1\Leftrightarrow \left(Nl_{i+1}=Nl_{i}\right)\wedge (Top_{i+1}=Top_{i})\wedge (Wop_{i}\neq 1)\wedge \;\;((Gn1_{i}-Gn1_{i+1}<0)\vee (Gn2_{i}-Gn2_{i+1}<0))$$
(3)$$N_{1}=1\Leftrightarrow Wn=1\;\wedge N_{3}=0\;\wedge (Gn1<32\vee Gn2<32)$$
(4)$$N_{2}=1\Leftrightarrow Wn=1\;\wedge \left(N_{1}+N_{3}=0\right)\wedge ((Gn1>33)\vee (Gn2>33))$$
(5)$$N_{3}=1\Leftrightarrow Nop=1\wedge \left(\left|Gn1-Gn2\right|\geq 2\right)$$
where:

*Wop*—replacement of the pantograph;

*Wn*—replacement of the pantograph sliding strip;

*Nl*—the locomotive number;

*Nop*—the pantograph number;

*Top*—the type of pantograph;

*Gn*1—thickness of the first carbon sliding strip;

*Gn*2—thickness of the second carbon sliding strip;

*N*_1_—replacement of the sliding strip due to the even wear of sliders;

*N*_2_—replacement of the sliding strip due to the detachment of a fragment of the sliding strip, material extraction or the burning of the sliding strip;

*N*_3_—replacement of the sliding strip due to the uneven wear of the sliders;

*i*—the measure number.

Developed algorithms, in turn, allowed the preparation of input data for machine learning. Learning data—predictors—are presented in Table 2.

### 2.4. Development of the Prediction Model

Machine learning is a currently used method in many fields. In this paper, the prediction model is based on algorithms for the classification of machine learning. For the development of the predictive model, we decided to use the MATLAB environment due to its strengths associated with machine learning. It also has a high-quality function library. The algorithms are compliant with industry standards, allowing to reduce the time required to develop solutions to the minimum. The tools used to validate the model are embedded in the application, enabling the developed model to be easily evaluated.

In order to develop the best predictive model based on the decision tree, the complex tree, medium tree and simple tree machine learning methods were tested.

Among the methods for classifying machine learning methods, the complex decision trees model proved to be the best. Figure 5 presents a graphic representation of this complex tree model. The tree in this form reflects how the classification decisions were made on the basis of attributes. The complex tree presented in Figure 5 was automatically made in the MATLAB software. The variables *x*_1_ ÷ *x*_10_ represent the input data used in the presented model:−*x*_1_—the review number;−*x*_2_—the identification of the new measure cycle;−*x*_3_—the number of days since the last replacement;−*x*_4_—the quarter of the year;−*x*_5_—the type of the pantograph;−*x*_6_—pantograph location (front, rear);−*x*_7_—the difference in thickness of the sliding strip N_1_ between technical review;−*x*_8_—the difference in thickness of the sliding strip N_2_ between technical review;−*x*_9_—the preview technical state;−*x*_10_—the replacement reason.

The decision tree is made up of decision nodes and leaf nodes. The text in the nodes depicts the optimal initial decision of splitting the tree based on *x_i_* values. The decision tree was built recursively. The root nodes were splatted recursively left and right until the maximum depth was reached. Each step in a prediction involves checking the value of one predictor (variable *x*_1_ ÷ *x*_10_) and calculating the Gini diversity index. In the proposed model, the maximum number of splits was 100; the Gini diversity index was applied as a split criterion, and there was no surrogate decision split. The Gini diversity index calculates the probability of a specific feature being incorrectly classified when randomly selected. If all the elements are linked to a single class, then it can be called pure. The Gini diversity index varies between values of 0 and 1, where 0 expresses the purity of classification, i.e., all the elements belong to a specified class or only one class exists there. Additionally, 1 indicates the random distribution of elements across various classes. In MATLAB software (MATLAB 2020b, MathWorks, Natick, MA, USA) it is possible to automatically generate the decision tree based on input data and determine the split criterion as well as the number of splits.

## 3. Results

### 3.1. Reliability Assessment of the Selected Types of Pantographs

Based on the technical reviews of locomotive types EU07 and EU09, the empirical data were collected which correspond to the failures of the selected pantograph types. The data were analyzed according to the Weibull analysis [IEC 61649:2008 Weibull analysis] and the parameters of 2-p Weibull distribution were obtained, as shown in Table 3. The Weibull parameters were calculated using the maximum likelihood estimation (MLE) method included in the Reliasfoft Weibull++ software, which allows taking into account the confidence bounds. The MLE method is preferred as the robust one in the case of a large data set. The probability density function of any probability mode can be expressed as [31]
(6)$$f\left(x;\theta_{1},\theta_{2},\ldots ,\theta_{k}\right)$$
where $\theta_{1},\theta_{2},\ldots ,\theta_{k}$ are the *k*-set of sought parameters related to *n* observations $x_{1},x_{2},\ldots ,x_{n}$ of the considered failure times of the pantograph. Then, the unknown parameters of any failure distribution can be obtained be maximizing the likelihood function given by
(7)$$L\left(\theta_{1},\theta_{2},\ldots ,\theta_{k}/x_{1},x_{2},\ldots ,x_{n}\right)=\overset{n}{\underset{i=1}{\prod }}f\left(x_{i};\theta_{1},\theta_{2},\ldots ,\theta_{k}\right)$$

The goodness of fit for the considered Weibull distribution with two-sided confidence bounds on reliability with a significance level of 0.05 for the selected pantograph types is shown in Figure 6a–c. The probability plot on the logarithmic grid shows the extent to which the Weibull distribution follows empirical data for the selected pantograph types. The significance level corresponding to the value of 0.05 adopted in the assessment determines the risk of committing the first type of error.

Probability density function for the Weibull distribution is given as follows [1]:(8)$$f\left(t\right)=\frac{\beta }{\eta^{\beta }}t^{\beta -1}exp\left[-{\left(\frac{t}{\eta }\right)}^{\beta }\right]$$

Based on the calculated parameters of a Weibull distribution, a probability density function may be plotted in order to compare the values of time at which the probability of the occurrence of failure reaches the maximum, *t*_(*f*)_ = max (Figure 6). This will be the basis for the further failure analysis and supervised machine learning.

**Figure 6 materials-14-05743-f006:**
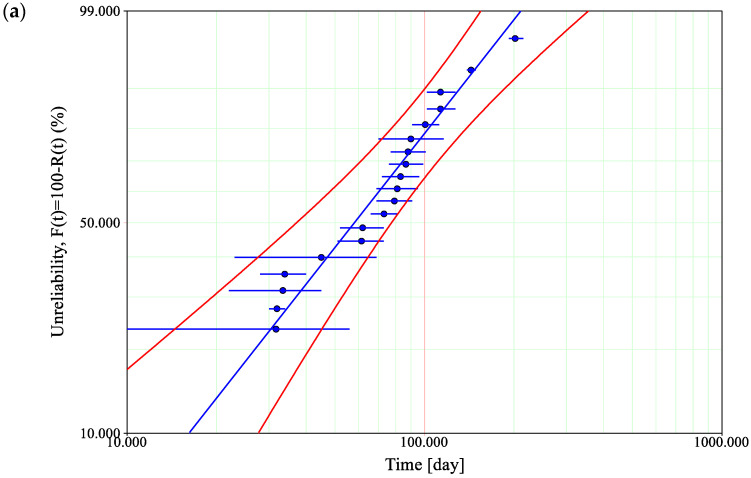
Weibull probability plots for the selected types of pantograph: (**a**) DSA-150; (**b**) AKP-4E; and (**c**) 5-ZL.

where:β−shape parameter
η−scale parameter

The obtained characteristics in Figure 7 refer to the probability density function of selected types of pantographs: DSA-150 (red); AKP-4E (black); and 5-ZL (blue). From a statistical point of view, they show the location of the determinant as a value that corresponds to the operating time value (*t_f_* = max) at which the probability of a given type of pantograph failure is greatest. Additionally, based on the probability density function, a mean time to failure may be calculated as follows:(9)$$MTTF=\int_{0}^{\infty }t\times f\left(t\right)dt$$

The values of MTTF and *t_f_* = max are presented in Table 4.

The obtained results indicate that the lowest value of MTTF is for the DSA-150 pantograph, which may suggest that it also has the lowest durability. However, it should be taken into account that the calculated MTTF values refer to the theoretical mean value of the time at which the failure may be observed, according to the approximation of the Weibull distribution. Therefore, the more important information is the value of operation time at which the failure occurrence is the most probable. In terms of practical purposes, such a value may indicate the actual durability of the pantograph. This approach may indicate the worst durability for the AKP-4 pantograph.

### 3.2. Complex Tree Errors Analysis 

A predictive model assumes classification into three different classes. Because the model based on the complex tree method gives the best results in terms of correct classification, the analysis of errors is only shown for the model presented in this paper. The analysis of errors in the assignment to different classes was made with the help of a confusion matrix. The matrix (3 × 3) in which the lines correspond to the correct decision classes and the columns correspond to the decisions predicted by the classifier is shown in Figure 8. At the intersection of the row *i* and column *j* is the number of examples originally belonging to the *i*-th class and included in the *j*-class.

As a result of the evaluation of the developed model, the correctness of the classification is approximately 81%. However, the correct classification of class 2 is the most important. This class indicates whether the sliding strip will soon fail (which defines class 3). Class 2 means that in the next time interval it will be necessary to replace the sliding strip. The prediction of this state thus makes it possible to reduce the number of failures to the sliding strips. This matrix is based on the input data represented by variables *x*_1_* ÷ x*_10_**.

Each value presented in the cell for the true class row and the predicted class column is calculated as follows:(10)$$CV=\frac{PC}{TC}\times 100\%$$
where:

*PC*—the number of true predicted cases;

*TC*—the number of true cases.

The true positive rate (*TPR*) is calculated as [32]
(11)$$TPR=\frac{TP}{P}=\frac{TP}{TP+FN}=1-FNR$$
where:

*TP*—true predictions of positive cases;

*P*—positive cases;

*FN*—false predictions of negative cases.

The false negative rate (*FNR*) is calculated as [32]
(12)$$FNR=\frac{FN}{P}=\frac{FN}{FN+TP}=1-TPR$$

In the confusion matrix, 49.6% of cases in the second class (2—indicating the limited possibility of their further use, as it will be necessary to replace the sliding strip for the next inspection) were identified as first class (1—possibility of further use). In practice, this means that approximately half of the cases can be identified before damage (before reaching class 3—no use, indicating that it is necessary to replace the sliding strip). Thus, a company can implement this method as predictive maintenance. This research is not finished, and different prediction models are still being analyzed. We are working with a hybrid model that uses classification and regression. Furthermore, other parameters such as weather and rail track which are used for rail vehicles are being analyzed. The initial research is promising, but completing this work will require more time.

### 3.3. Failure Analysis

The failure analysis included two variants. The reference variant was only based on processed archival data. For the analysis of variant I, data from technical reviews modified by the predictive model were used. The structure determined during the preliminary data processing in step 1 was also used in step II, thanks to which it was possible to compare the results.

Figure 9 shows the correctness of the classification of the technical states of a pantograph. The meanings behind the three technical states used are as follows:

1—possibility of further use;

2—limited possibility of further use and it will be necessary to replace the sliding strip for the next inspection;

3—no use and it is necessary to replace the sliding strip.

The test results concerning the correct and incorrect classification of technical conditions, as well as the number of cases for which it was possible to predict the appropriate technical condition, are presented in Table 5. 

The technical condition of a pantograph is correctly classified when the technical condition is the same as the real one (based on archival data):(13)$$c_{c1}=1\;\Leftrightarrow p_{c}=1\;\wedge p_{c}=r_{c}$$
(14)$$c_{c2}=1\;\Leftrightarrow p_{c}=2\;\wedge p_{c}=r_{c}$$
(15)$$c_{c3}=1\;\Leftrightarrow p_{c}=3\;\wedge p_{c}=r_{c}$$
where:

$c_{cn}$—correctly classified technical conditions n;

$p_{c}$—predicted technical condition;

$r_{c}$—real technical condition (from archival data).

On the other hand, the number of anticipated cases was calculated by checking whether the actual technical condition was correctly predicted. By analyzing the data in the reference variant, there were 47 failures to the sliding strip or to the current collector. In variant I (after machine learning), only 23 failures were noted. The use of the presented model allows reducing the number of failures by approximately 50%.

## 4. Discussion

Looking at the results which were obtained using a decision tree, it can be seen that this enables properly categorizing defined technical conditions to varying degrees. When analyzing the results of this research, it can be noted that in the case of predicted state 3, 2.08% of cases were classified to this state despite the fact that they actually belonged to a different technical state. However, all cases which were actually in state 3 were correctly predicted (100%). This means that in the case of this condition, 2.08% of cases were additionally incorrectly classified under this technical condition.

To achieve the presented results, wider research was performed. In this research, different classification methods were used; however, the best result was achieved by using the complex tree classification model. The other analyzed methods include decision trees (medium tree, simple tree); supporting vector machines (SVMs) (linear, quadratic, cubic, fine Gaussian, medium Gaussian, coarse Gaussian); the metric of nearest neighbors—(KNN) (fine, medium, coarse, cosine, cubic, weighted); classifiers (boosted trees, bagged trees, subspace discriminant, subspace KNN, RUS boosted trees). Among the tested methods of classification, the decision trees were proven to obtain the best results for a formulated problem. Complex trees obtained results which were between 0.5 and 4% better than other machine learning classification methods in terms of the correctness of the classification of technical conditions.

The most important in terms of reducing damage to the current collectors is state 2, which means that it will soon be necessary to replace the sliding strip or the entire collector. In the case of the decision-making machine learning method proposed in this article, it can be noted that it achieves approximately 50% correct identification of this state, and also correctly categorizes 50% of actual cases into this state. 

Therefore, this means that in 50% of cases, it will be possible to detect the need for replacement before damage occurs or the limit value of the sliding strip thickness will be exceeded. Studies have shown that the use of artificial intelligence methods can be successfully applied in the preventive diagnostics of rail vehicles. However, it is necessary to define appropriate analytical and simulation methods, as well as diagnostic procedures that enable the detection of as many cases as possible. There is also a need to project a method which can support appropriate maintenance activities in order to minimize the number of failures. 

## 5. Conclusions

In conclusion, the presented methodology is based on the artificial intelligence decision tree. Failure analysis is necessary for correctly preparing the input data for both variants. The reference variant only concerns the analysis of data obtained during the technical review when variant I is based on the classification machine learning method developed herein.

The results show that the prediction of a technical condition can reduce the number of sliding strip failures by approximately 50%. Therefore, the costs related to the repair of damaged railway infrastructure caused by the poor technical condition of pantograph collectors can be significantly reduced. The application of the developed method would also enable the reduction in railway delays caused by the damage of the current collector system—the traction network.

Further research will focus on the development of a predictive model that allows predicting damage with 100% accuracy so that negative effects can be eliminated.

## Figures and Tables

**Figure 1 materials-14-05743-f001:**
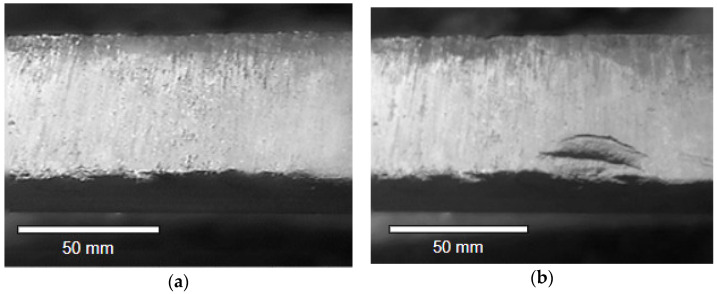
Damages of the edge of the carbon sliding strip: (**a**) minor surface damages; and (**b**) major surface damages.

**Figure 2 materials-14-05743-f002:**
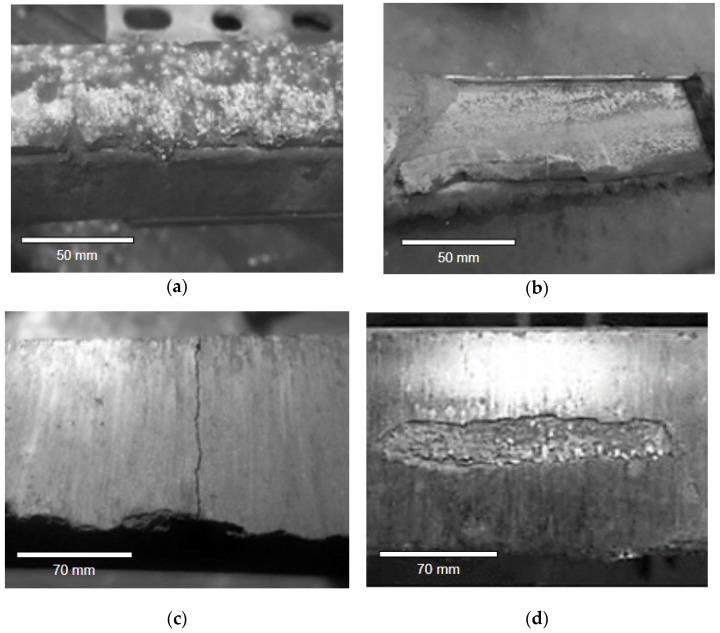
Wear of the sliding: (**a**) material melting as a result of arcing; (**b**) detachment of a piece of carbon strip; (**c**) crack of a strip; and (**d**) the top layer of a strip is peeling off [30].

**Figure 3 materials-14-05743-f003:**
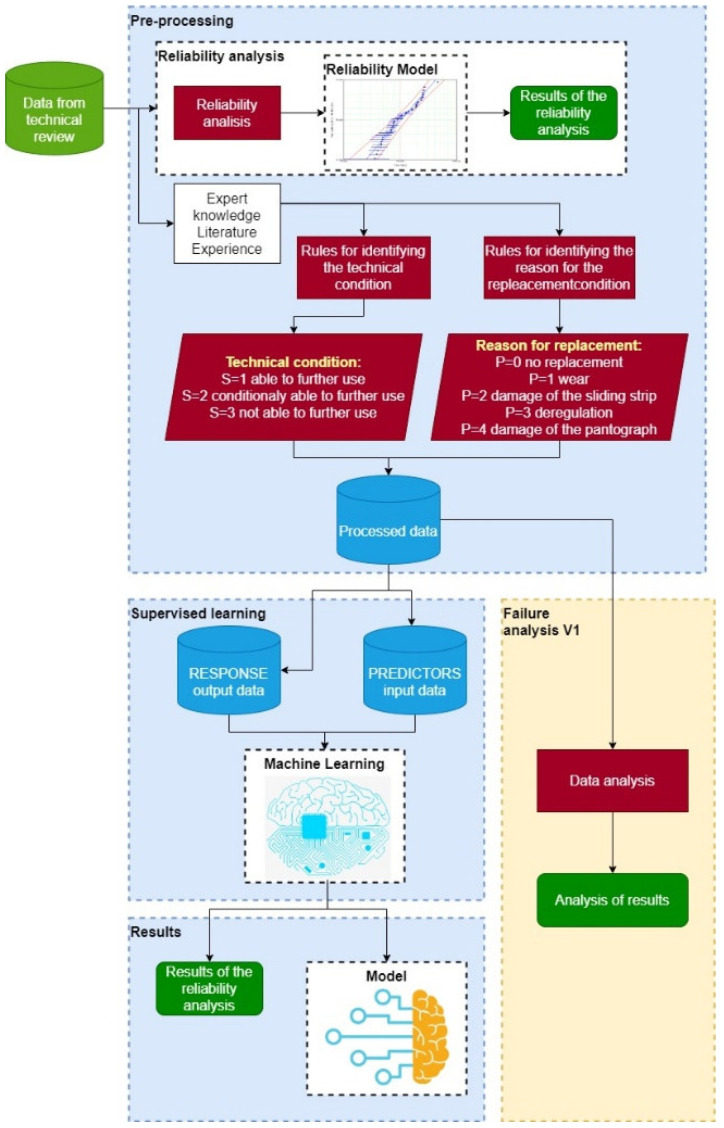
Method of three-valued prediction of technical condition based on machine learning and failure analysis for the reference variant—step 1.

**Figure 4 materials-14-05743-f004:**
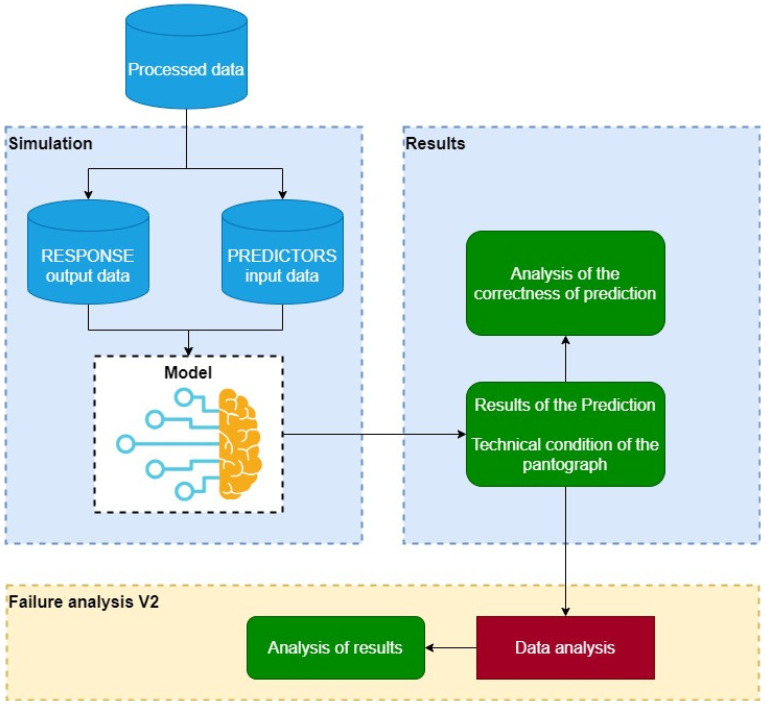
Method of three-valued prediction of technical condition based on machine learning and failure analysis for variant I—step II.

**Figure 5 materials-14-05743-f005:**
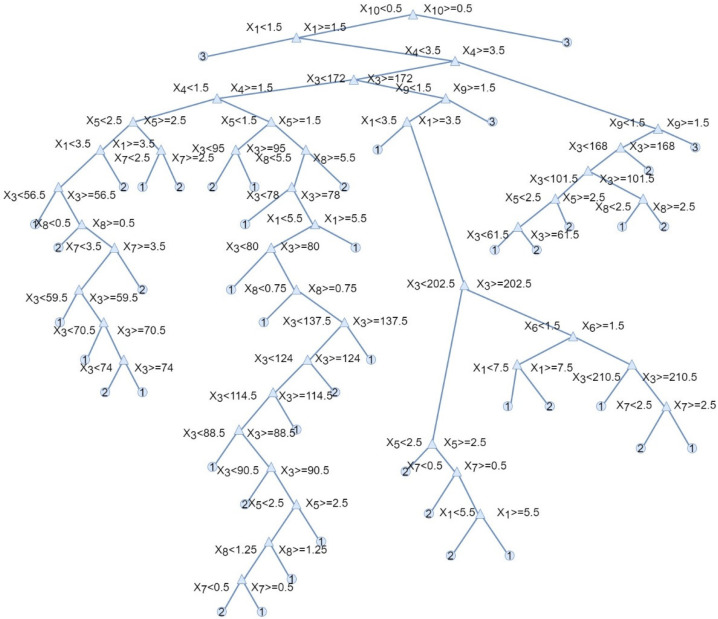
Complex tree.

**Figure 7 materials-14-05743-f007:**
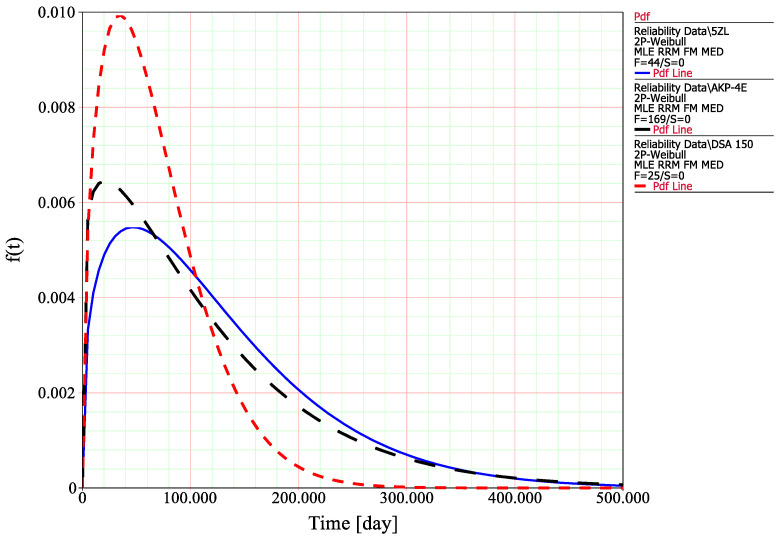
Probability density function for the selected types of pantograph: (**a**) DSA-150 (red); (**b**) AKP-4E (black); and (**c**) 5-ZL (blue).

**Figure 8 materials-14-05743-f008:**
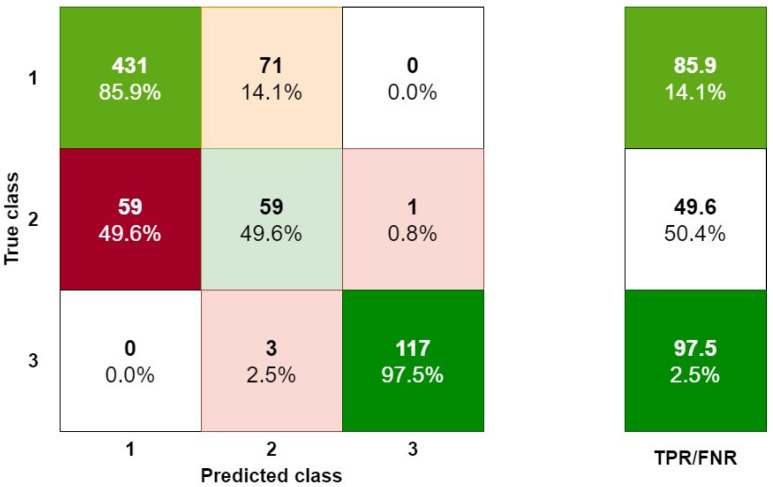
Confusion matrix for the decision tree.

**Figure 9 materials-14-05743-f009:**
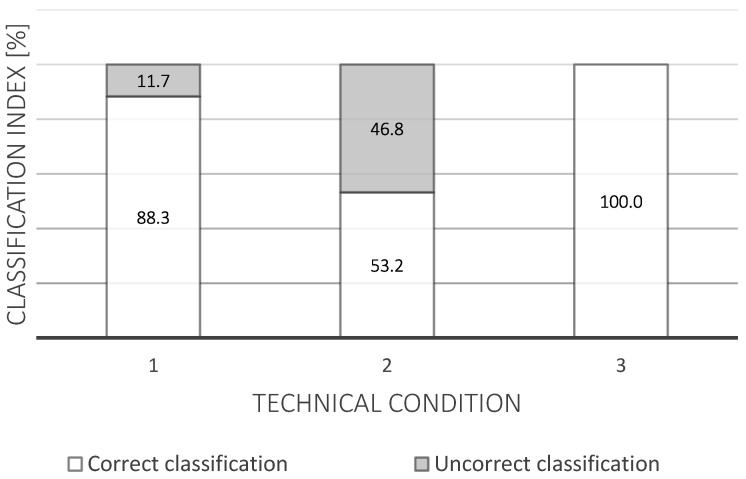
Correctness of classification of the technical condition of a pantograph.

**Table 1 materials-14-05743-t001:** Technical data of carbon sliding strips for different sample manufacturers [29].

	Manufacturer	Type	Properties			
No.			Resistance	Density	Hardness	Flexural Strength
			(µohm)	(g/cm^3^)	(HRB)	(N/mm^2^)
1	Morganite	MY7A2	5	2.40	-	85
2	PanTrac GmbH	RH83M6	7	3.40	105	102
3	Elektrokarbon a.s.	SK181	-	2.20	90	-
4	Mersen (France)	P5696	7	2.30	90	85
5	Required value	-	Max 5	Max 2.50	Max 120	Min 65

**Table 2 materials-14-05743-t002:** Machine learning predictors.

Description	Symbol
Review number	i
A new measuring cycle	Cnew
The number of days since the replacement	*D*
The quarter of the year	Q
Current collector type	*Top*
Front/rear current collector	Cc
Difference in the N_1_ thickness between reviews	Th1
Difference in the N_2_ thickness between reviews	Th2
Earlier technical condition	S
The reason for the replacement	N

**Table 3 materials-14-05743-t003:** Parameters of Weibull distribution for the analyzed types of pantograph.

Parameters of Weibull Distribution	DSA-150	AKP-4E	5-ZL
*β*	1.470337	1.153632	1.329664
*η* (days)	74.619087	119.655763	134.032580

**Table 4 materials-14-05743-t004:** Calculated MTTF and *t_f_* = max parameters for the selected types of pantographs.

Type of Pantograph	MTTF (Day)	*t_f_* = max (Day)
DSA-150	67.53	35
AKP-4	113.77	21
5-ZL	123.25	48

**Table 5 materials-14-05743-t005:** Results of a technical condition prediction.

The Number of Predicted Technical Conditions	Technical Condition			
	All	1	2	3
Correctly classified technical conditions (%)	84.62	89.16	52.08	97.92
Incorrectly classified technical conditions (%)	15.38	10.84	47.92	2.08
Predicted cases (%)	84.62	88.29	53.19	100.00
Unpredicted cases (%)	15.38	11.70	46.81	0.00

## Data Availability

Not applicable.

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
