# Peer review of "Pantograph Sliding Strips Failure—Reliability Assessment and Damage Reduction Method Based on Decision Tree Model"

_materials, 2021, doi:10.3390/ma14195743_

Round 1

Reviewer 1 Report

This paper is to develop the best predictive model based on decision tree. It is an interesting paper. However, the writing about decision tree and some statistics are very sloppy. 1 Where did the exemplary algorithms come from? 2 What is the Ginis Diversity Index? 3 What is Confusion Matrix for Decision Tree? Please give us a mathematical definition. Please refer to some books like "Fundamentals of Machine Learning for Predictive Data Analytics Algorithms, Worked Examples, and Case Studies" By John D. Kelleher, Brian Mac Namee and Aoife D'Arcy and then provide more details about how to construct decision trees? Also what is Maximum Likelihood Estimation? What is "Goodness of fit for the considered Weibull distribution with two-sided confidence bounds on reliability with significance level of 0.05"? Please refer to some books on statistical inference and give us more math details.

Author Response

Dear Reviewer, thank you very much for your valuable comments and suggestions. We have revised the paper accordingly to them. Detailed responses to the comments and suggestions are as follows.

Comment 1. Where did the exemplary algorithms come from?

Answer 1. Presented in Figures 3 and 4 algorithms were developed by us. It is the full results of our research. The sub-algorithms like Complex Tree used as a part of the main algorithms presented in Figures 3 and 4 come from the software Matlab.

Comment 2. What is the Ginis Diversity Index?

Answer 2. Thank you for paying attention to the lack of a definition of the indicator. Gini Index calculates the amount of probability of a specific feature that is classified incorrectly when selected randomly. If all the elements are linked with a single class then it can be called pure. The Gini index varies between values 0 and 1, where 0 expresses the purity of classification, i.e. all the elements belong to a specified class or only one class exists there. And 1 indicates the random distribution of elements across various classes. We add this description to the paper before Figure 5.

Comment 3. What is Confusion Matrix for Decision Tree? Please give us a mathematical definition. Please refer to some books like "Fundamentals of Machine Learning for Predictive Data Analytics Algorithms, Worked Examples, and Case Studies" By John D. Kelleher, Brian Mac Namee and Aoife D'Arcy and then provide more details about how to construct decision trees?

Answer 3. We add the appropriate formulas before Figure 8 with regards to the TPR and FNR definition presented in the suggested book. We also provide more details about the methodology of a construct decision tree.

Comment 4. Also, what is Maximum Likelihood Estimation? What is "Goodness of fit for the considered Weibull distribution with two-sided confidence bounds on reliability with a significance level of 0.05"? Please refer to some books on statistical inference and give us more math details.

Answer 4. Thank you for any inaccuracy found. MLE method is preferred as the robust one for calculating the parameters of probability distribution in case of large data set. The probability plot on the logarithmic grid shows to what extent the Weibull distribution follows the empirical data for the selected types of pantograph. The significance level corresponding to the value of 0.05 adopted in the assessment determines the risk of committing the first type of error. These sentences as well as the relevant equations and references were added in section 3 of the manuscript.

Dear Reviewer, thank you very much for your time, professionalism and advices.

Reviewer 2 Report

The manuscript represents an application of data-driven methods in materials sciences. Authors propose decision tree (DT) for reliability assessment and damage reduction in Pantograph sliding strips failure. On of the major comments would be why the DT has been selected for this application how DT performs in a comparative analysis with other machine learning methods. The other matter to be discussed is that how the performance of the DT for reliability assessment and damage reduction can be improved? how ensembles and hybrid methods can help to improve the efficiency, performance, computation costs, and reliability. Such matters are to be elaborated in the manuscript. The other issue is with the validation. Further elaboration on the validation and limitation of the study would be essential. For the future works authors described that the applied reliability models of wear propagation can be extended by the parameters of the cost and repair time becoming the basis for estimating the costs of operation and maintenance, please elaborate this as a limitation of the study and include a paragraph in the introduction.

The application has significance as the the damage of the pantograph or sliding strip may cause blocking the railway line. Thus model accuracy and validation would be essential and important. Thus selecting the the evaluation metrics are performance of the method must be elaborated and discussed.

Please give several references on "preventive maintenance strategy related to the pantograph" described in the introduction.

Please provide refernces to the technical data of carbon sliding strips for sample manufacturer.

In figure 1, instead of using machine learning it is perhaps better to name the method.

Enlarge figure 7.

In the dicsussion the limitation and lack of the study on including a comprative analysis is missing. Describe why not including several other ML methods.

Author Response

Dear Reviewer, thank you very much for your valuable comments and suggestions. We have revised the paper accordingly to them. Detailed responses to the comments and suggestions are as follows.

Comment 1. Please give several references on "preventive maintenance strategy related to the pantograph" described in the introduction.

Answer 1. We added the description of previous works done by other scientists related to pantograph predictive/preventive maintenance to the introduction.

Comment 2. Please provide references to the technical data of carbon sliding strips for sample manufacturer.

Answer 2. The technical data of carbon sliding strips for sample manufacturer presented in Table 1 comes from the symposium organized by the polish Railway Institute (Instytut Kolejnictwa). In the presentation “Applications of carbon overlays in current collectors” W.Majewski present the technical data of carbon sliding strips used in polish railways. We add the reference to Table 1.

Comment 3. In figure 1, instead of using machine learning it is perhaps better to name the method.

Answer 3. We suppose it was about Figures 3 and 4 (Machine learning is presented in those figures), in figure 1 are presented damages. We call our method a Method of three-valued prediction of technical conditions based on machine learning and failure analysis and add this to the title of Figures 3 and 4.

Comment 4. Enlarge figure 7.

Answer 4. Thank you for your suggestion. I would like to inform that Figure 7 has been adapted to the requirements of the Journal. Elements of the Figure have been enlarged to increase its readability and clarity. Additionally, this change has also been applied to Figures 6a-6c.

Comment 5. In the discussion, the limitation and lack of the study on including a comparative analysis are missing. Describe why not including several other ML methods.

Answer 5. Thank you for finding our imperfection. In our research, we tested different classification methods. Complex trees gave results from 0.5 to 4% better for the correctness of the classification of technical conditions than other machine learning classification methods. In a discussion, we add information about different analyzed methods.

Dear Reviewer, thank you very much for your time, professionalism and advices.

Reviewer 3 Report

The article submitted for review presents the identification of slide damage in a very interesting way. However, it needs a few improvements to make it more attractive to the reader.
Figure 1 shows minor damage - it is worth pointing this out as it is difficult to see as it stands. It is also worth describing more clearly in the text what the minor damage is.
In the caption of figure 2 there is: c)crack of a strip, it should be: c) crack of a strip - spaces are missing
It is worth describing the variables contained in relations 1 - 5 more clearly
Wiebull's relationship should also be clarified as to what the variables mean.
In figure 6 I suggest enlarging the axis captions, in relation to the graphs they are not very readable.
In Figure 7, the legend is illegible.
Also figure 7 needs clarification in the text as to what the individual variables mean.
The number of references from the same author should be verified according to the requirements of the journal.

Author Response

Dear Reviewer, thank you very much for your valuable comments and suggestions. We have revised the paper accordingly to them. Detailed responses to the comments and suggestions are as follows.

Comment 1. Figure 1 shows minor damage - it is worth pointing this out as it is difficult to see as it stands. It is also worth describing more clearly in the text what the minor damage is.

Answer 1.  In Figure 1a the minor damages are presented. These damages is following from the uneven edge of the sliding strip. In Figure 2b the uneven edge and material extraction are presented. The uneven edge of the pantograph sliding strips may wear the overhead contact line more quickly or even damage it.

Comment 2. In the caption of figure 2 there is: c)crack of a strip, it should be: c) crack of a strip - spaces are missing.

Answer 2. Thank you for your insight. We add the missing space.

Comment 3. It is worth describing the variables contained in relations 1 - 5 more clearly Wiebull's relationship should also be clarified as to what the variables mean. In figure 6 I suggest enlarging the axis captions, in relation to the graphs they are not very readable.

Answer 3. Thank you noticed imperfections. We add the descriptions and explain the variables used in equations 1-5. We also normalized the variables name used in the equations. We provided additional explanations to the variables and related Maximum Likelihood Estimation (MLE) method used.  MLE method is preferred as the robust one for calculating the parameters of probability distribution in case of large data set. The probability plot on the logarithmic grid shows to what extent the Weibull distribution follows the empirical data for the selected types pantograph. The significance level corresponding to the value of 0.05 adopted in the assessment determines the risk of committing the first type of error. These sentences as well as the relevant equations and references were added in section 3 of the manuscript.

Comment 4. In Figure 7, the legend is illegible. Also figure 7 needs clarification in the text as to what the individual variables mean. The number of references from the same author should be verified according to the requirements of the journal.

Answer 4. Thank you for your suggestion. I would like to inform that Figure 7 has been adapted to the requirements of the Journal. Elements of the Figure have been enlarged to increase its readability and clarity. Additionally, this change has also been applied to the Figures 6a-6c. Explanations to Figure 7 relating to individual variables have been also added in Chapter 3.

Dear Reviewer, thank you very much for your time, professionalism and advices.

Reviewer 4 Report

Dear Authors,

It will be very fruitful to develop an accurate predictive model. Several things seem not clearly described. Detailed questions are as follows.

1)    [2.2] What do you mean by the archival data? Are they review times and damages shown in Figures 1 and 2? Or other operation data?
2)    [2.3] Several symbols in Equations (1)-(5) are used without any description. Please, insert the descriptions and explain the equations.
3)    [2.4] Was the complex tree in Figure 5 is constructed automatically or manually? And what do X1, X2, … in complex tree indicate?
4)    [3.1] What is the cause of different probability functions in Figure 7 for the selected types of pantographs? Please, compare the damage types of the selected types of pantographs.
5)    [3.2] Three pantograph types were analyzed. The confusion matrix in Figure 8 is still for all three pantograph types? Or a part of them?
6)    [3.2] In Figure 8, 59 class 2 cases were incorrectly predicted as class 1 cases. Nearly a half of “limited possibility of further use” cases were predicted as “possibility of further use” cases. Maybe it cannot be allowed for safety concerns. What is the cause of such inaccurate predictions? Can it be solved?

Author Response

Dear Reviewer, thank you very much for your valuable comments and suggestions. We have revised the paper accordingly to them. Detailed responses to the comments and suggestions are as follows.

Comment 1. [2.2] What do you mean by the archival data? Are they review times and damages shown in Figures 1 and 2? Or other operation data?

Answer 1. Archival data means that this data is not actual - comes from past technical reviews. The presented method also can be used in real-time. So if the company will use this method the results are based on the on-time data. The photos presented damages in figures 1 and 2 were taken during the technical reviews and after the failure of the pantograph during transportation.

Comment 2. [2.3] Several symbols in Equations (1)-(5) are used without any description. Please, insert the descriptions and explain the equations.

Answer 2.  Thank you noticed imperfections. We add the descriptions and explain the variables used in the equations 1-5. We also normalized the variables name used in the equations.

Comment 3. [2.4] Was the complex tree in Figure 5 is constructed automatically or manually? And what do X1, X2, … in complex tree indicate?

Answer 3.  The Complex Tree presented in Figure 5 was made automatically in the Matlab software. We add the description of the variables x1÷x10 before the Complex Tree. This variables represents the input data uses in the presented model.

Comment 4. [3.1] What is the cause of different probability functions in Figure 7 for the selected types of pantographs? Please, compare the damage types of the selected types of pantographs.

Answer 4. The considered types of pantographs with sliding strips were operated in the same conditions and were subject to the same wear processes. The causes of failure of pantographs and their sliding strips are common as described in chapter 2. Therefore, the registered failure cases of pantographs are purely random (stochastic).

Comment 5. [3.2] Three pantograph types were analyzed. The confusion matrix in Figure 8 is still for all three pantograph types? Or a part of them?

Answer 5. The confusion matrix in Figure 8 is for all three pantograph types. This matrix base on the input data represented as variables x1÷x10, and x5 variables describe the type of pantograph. We add these comments before Figure 8.

Comment 6. [3.2] In Figure 8, 59 class 2 cases were incorrectly predicted as class 1 cases. Nearly a half of “limited possibility of further use” cases were predicted as “possibility of further use” cases. Maybe it cannot be allowed for safety concerns. What is the cause of such inaccurate predictions? Can it be solved?

Answer 6. After the Confusion matrix in Figure 8, we add the comments to the wrong classified 2 cases. Despite this problem the presented method is useful, because inspection) were identified as first class (1 - the possibility of further use). In practice, it means that about half of cases can be identified before the damage. The promising are the hybrid methods, and using the working rail track and weather as additive parameters – we are analyzing them in our research, but this research is now at the begging.

Dear Reviewer, thank you very much for your time, professionalism and advices.

Round 2

Reviewer 1 Report

The revised version can be accepted.  

Reviewer 4 Report

Dear Authors,

Thank you for responding to all review comments well.